# The Novel Application of Geometric Morphometrics with Principal Component Analysis to Existing G Protein-Coupled Receptor (GPCR) Structures

**DOI:** 10.3390/ph14100953

**Published:** 2021-09-23

**Authors:** Daniel N. Wiseman, Nikita Samra, María Monserrat Román Lara, Samantha C. Penrice, Alan D. Goddard

**Affiliations:** 1School of Biosciences, College of Health and Life Sciences, Aston University, Birmingham B4 7ET, UK; d.wiseman1@aston.ac.uk (D.N.W.); 180080476@aston.ac.uk (N.S.); 170230018@aston.ac.uk (M.M.R.L.); 2School of Technology, BPP University, BPP House, Aldine Place, 142-144 Uxbridge Road, London W12 8AA, UK; SamPenrice@bpp.com

**Keywords:** GPCR, ligand, activity, thermostabilised, recombinant, structure, geometric morphometrics, principal component analysis

## Abstract

The G protein-coupled receptor (GPCR) superfamily is a large group of membrane proteins which, because of their vast involvement in cell signalling pathways, are implicated in a plethora of disease states and are therefore considered to be key drug targets. Despite advances in techniques to study these receptors, current prophylaxis is often limited due to the challenging nature of their dynamic, complex structures. Greater knowledge and understanding of their intricate structural rearrangements will therefore undoubtedly aid structure-based drug design against GPCRs. Disciplines such as anthropology and palaeontology often use geometric morphometrics to measure variation between shapes and we have therefore applied this technique to analyse GPCR structures in a three-dimensional manner, using principal component analysis. Our aim was to create a novel system able to discriminate between GPCR structures and discover variation between them, correlated with a variety of receptor characteristics. This was conducted by assessing shape changes at the extra- and intracellular faces of the transmembrane helix bundle, analysing the XYZ coordinates of the amino acids at those positions. We have demonstrated that GPCR structures can be classified based on characteristics such as activation state, bound ligands and fusion proteins, with the most significant results focussed at the intracellular face. Conversely, our analyses provide evidence that thermostabilising mutations do not cause significant differences when compared to non-mutated GPCRs. We believe that this is the first time geometric morphometrics has been applied to membrane proteins on this scale, and believe it can be used as a future tool in sense-checking newly resolved structures and planning experimental design.

## 1. Introduction

G protein-coupled receptors (GPCRs) are one of the most widely studied families of membrane proteins in the human genome due to their extensive involvement in a plethora of cell signaling pathways [1]. These seven-transmembrane eukaryotic receptors contribute to the normal function of a cell but can also be responsible for a widespread variety of disease states. While approximately 40% of all drugs target these receptors [2], this must be contrasted with a relatively low coverage of the GPCR family overall (around 19%), leaving many conditions with little to no prophylactic options [3]. Advances in drug design and discovery are therefore needed to expand this coverage; further understanding of the structure–function relationship of GPCRs will aid with the development of orthosteric and allosteric ligands against these receptors, widening the pool of potentially druggable GPCRs [4]. This paper therefore presents a novel application of a mathematical technique, geometric morphometrics with principal component analysis, to further enhance the field’s knowledge and understanding of GPCR structures.

GPCRs are typically composed of three domains as shown in Figure 1; the N terminus which is extracellular, the transmembrane (TM) domain and the C terminus which is intracellular. These are connected by three extra- and three intracellular loops [5]. While the TM domain always contains seven alpha-helices and the C terminus is relatively short, the N terminus varies in size depending on the classification of the receptor. As the N terminus can form a part of the location of orthosteric ligand binding, the ligands also vary in size and characteristic, ranging between ions and proteins [6]. Referring to the original system of classification, there are six main GPCR families (named A–F), as summarised in Table 1. Additionally, the alternative GRAFS system describes the main five vertebrate GPCR families: (G)lutamate, (R)hodopsin, (A)dhesion, (F)rizzled/Taste2 and (S)ecretin [7]. This paper will continue to use the classical system, alongside naming the appropriate receptor sub-groups (for example, the neurotensin receptors within family A).

Following binding of the extracellular ligand, the receptor undergoes a characteristic conformational change, promoting a signalling cascade within the cell [9]. This movement is proposed to include a series of molecular switching and repacking of key contacts within the intra- and interhelical arrangement, followed by the outward movement of TMs 5 and 6 away from the TM bundle [10]. TM6 is then thought to undergo a mechanical rotation at the intracellular face, allowing exposure of a binding cleft which enables recruitment of the G protein, and subsequent propagation of the signalling cascade. Movement is therefore thought to be focussed around the intracellular end of TM6 and more recently, TM7 has also been observed to move towards TM3 [11]; overall, the molecular activation of GPCRs remains complicated to elucidate exactly. As shown in Figure 1b, heterotrimeric G proteins are composed of three subunits (α, β and γ); the α subunit facilitates an exchange of GDP for GTP, while the βγ complex splits away to signal independently from Gα [9]. As humans encode 18, 5 and 12 different α, β and γ subunits respectively, these combine into a variety of stimulatory or inhibitory effects on the subsequent signalling pathways. These intracellular signals often result in changes to protein activity or transcription factors, affecting cell behaviour (second messenger signaling, for example) depending on the combination of ligand and receptor in question [9]. Nevertheless, the exact molecular determinants underlying GPCR-G protein coupling also remain to be fully elucidated.

Overall, the relationship between structure and function of GPCRs is of utmost importance as the tertiary structure of the receptor contributes to determining the orthosteric and allosteric ligand binding domains, the efficiency of the conformational change and the receptors’ interaction with other proteins and lipids as well [12]. It would therefore seem logical to conclude that intended or unintended modifications of a GPCRs wildtype sequence/structure may impact on its efficacy as a functional membrane protein. Despite this, researchers frequently perform such modifications including thermostabilising mutations and chimeric fusions because of the need for enhanced stability during expression and purification, for example [13,14,15,16].

We therefore hypothesised that resolved structures would not only show differences between conformational states, but that any modifications may also show a difference when compared to the unmodified structure. Any significant differences may therefore correlate with constraining the receptor to a certain conformation, of which the researcher should be aware. However, equally, no differences observed may indicate that the particular modification is ‘safe’ to use, and the GPCR structure is not significantly impacted upon. To study these potential differences between resolved GPCR structures, the design rationale highlighted the importance of an objective and robust system of collecting data, with unbiased and mathematical analyses.

Geometric morphometrics was selected for this task as it can quantify variation between shapes and fulfils the criteria of the design rationale; when the data is transformed, a morphospace is produced where objects are positioned based upon differences or similarities in shape [17,18,19]. Geometric morphometrics can therefore quantifiably capture and display these shape variations between resolved GPCR structures, in a three-dimensional format. The data required for this method are Cartesian landmark coordinates and should be common reference points between the shapes [18]. As such, the seven transmembrane helices of GPCRs can be used as they are the defining structural characteristic of these receptors and remain relatively conserved between receptors. In order to assess any shape variation at the extra- and intracellular face of the receptor, both ends of the TM helices were selected as reference points; each receptor, by definition, contains these landmarks. More specifically, the XYZ coordinates of the alpha-carbon atom (Cα) of the amino acid residue at each end of the helix (meaning the first and last residue of each TM helix) were selected as the Cartesian landmarks to minimize variation due to amino acid identity at these positions.

A comprehensive explanation of geometric morphometrics can be found elsewhere [17,18,20]; however, the main elements are as follows. The landmark coordinates are first size- and rotation-adjusted by a Procrustese superimposition; this is an orthogonal transformation to standardise and scale the data for comparison [21]. Subsequently, a covariance matrix is generated from which principal component analysis is then performed. Principal components are a series of vectors which show variation in the data and are composed of the eigenvectors of the data’s covariance matrix. An eigenvector is a direction with variation, and its eigenvalue is the variance of that direction [22]. As such, the principal components of the data produce a morphospace where the first principal component is responsible for the greatest variation between the coordinate data [23]. As principal component analysis uses covariance, its analysis is two-dimensional and therefore principal component scores can be compared; a comparison between principle components 1 and 2 (PC1 and PC2) will show the greatest variation in the morphospace. Overall, geometric morphometrics with principal component analysis breaks down three-dimensional landmark coordinates to find patterns in the data, because of their shape variations, and can be further analysed statistically by analysis of similarity (ANOSIM) and permutational multivariate analysis of variance (PERMANOVA) [17,23].

We therefore present an application of geometric morphometric analysis to resolved GPCR structures which is, to our knowledge, the first demonstration of its suitability for membrane protein analysis. Overall, we have been able to classify, discriminate and mathematically determine significant differences between GPCR structures based on their characteristics such as activation state, bound ligands and fusion proteins. While this is a method more often used in disciplines such as anthropology and palaeontology, we believe that this technique has wide applicability to the structural biology field. By increasing understanding of GPCR structures resolved under varied conditions, it is hoped that intelligent, structure-based ligand design will continue to improve and expand the current range of druggable receptors.

## 2. Results

### 2.1. From XYZ Coordinates to Principal Component Analysis

Before detailing some examples of the receptor group analyses, the steps leading to the final principal component analysis should be highlighted first for context. This should also familiarise the reader with the various figures produced by the MorphoJ software [24]. The extracellular face of the β2-adrenergic receptors (β2AR) is shown as the example in Figure 2, though each subsequent analysis presented always entailed the following process.

Initially, after importing the appropriate XYZ coordinate data from PDB files into the MorphoJ software, a generalised Procrustese transformation was performed with alignment to the principal axes. To effectively compare the shape variation, the data must first be standardised. The Procrustese transformation does this by translation, rotation, scaling and superimposition of the coordinate data (Figure 2a). The blue numbered circles are the average of the positions included in the dataset (in this case, of the extracellular face of the β2AR TM helix bundle), and the smaller black dots around them are each individual structure’s position. Once the selected dataset had been standardised, a covariance matrix was generated; this measures how much shapes vary together and allows for statistical analyses—in this case, principal component analysis.

Firstly, the principal component (PC) scores are calculated through eigendecomposition of the covariance matrix values and projection of shape variables onto low-dimensional space (a 2D graph, for example). Figure 2b shows the overall direction and magnitude of the greatest variation caused by PC1 of this dataset; to interpret the lollipop graph, the blue circle is the average position of each helix, while the protruding stick represents its overall movement. Figure 2b therefore displays an equal movement of TM6 away from the helix bundle, and TM1 towards TM2; conversely, for this dataset, TMs 2, 5 and 7 move very little in comparison.

Principal components are directional vectors which maximise data variation and are ranked by the percentage variation they cause, based on the dataset’s eigenvectors and eigenvalues (Figure 2c). PC1 will always represent the greatest variation, with PC2 the second most, and so on. As shown in Figure 2c, the first two principal components combined account for more than 70% of the shape variations in the structures in this dataset. The greatest comparison of variation can therefore be observed by comparing PC1 and PC2 (Figure 2d). At this point, if two receptor structures were identical, they would occupy the same space on this graph. However, there are clear differences between the structures in this dataset and correlation with characteristics need to be inferred manually. This is because principle component analysis can show variation between shapes in an unbiased manner but cannot explain or interpret the data. From here on, examples of receptor group analyses will be shown comparing both the extra- and intracellular data (left and right, respectively). The comparisons between PC1 and PC2 will be shown in text, along with the results of the statistical tests; further relevant figures can also be found in the indicated appendices.

### 2.2. Activation State—The β2-Adrenergic Receptors

GPCR activation largely relies upon ligand binding, and so this was the first analysis performed; additionally, the active, intermediate and inactive states are well characterised within the available structures, providing a proof of concept for geometric morphometrics in this context. As summarised earlier, receptor activation entails movement of the TM bundle, and rotation of TM6 to enable recruitment of G proteins and other accessory proteins to the intracellular face [10]. It was therefore hypothesised that GPCR structures captured in different activation states would show variation between each other, particularly at the intracellular face where movement is focussed. The β2-adrenergic receptors (β2AR) were selected to highlight this example, not only because it is a widely studied GPCR, but this group had ample numbers to analyse, and a mixture of activation states based on the hypothesis. Of the 36 β2AR structures studied, 23 were described as inactive, 7 as active and 6 as active bound to G_s_. A list of the PDBs is provided in Appendix A
Table A1. Inactive structure 5JQH was not included in analyses due to heavily skewing the data [25]; we hypothsise that it is possibly due to it being allosterically nanobody stabilised.

Once the PC scores were generated, colours were assigned to each activation state as defined by the GPCRdb database [26], where inactive is red, active is blue and active with G_s_ coupled is green (Figure 3). The β2AR extracellular face data in Figure 3a shows both active and inactive categories generally occupying the same morphospace suggesting that when these receptors were captured in their states, the extracellular face of the TM helices do not discriminate between inactive and active. ANOSIM and PERMANOVA were then carried out on the principal component scores to test for the significance of clusters and distance. These gave relatively low R and F values overall (0.2965 and 4.142, *p* = 0.0047 and *p* = 0.0003), indicating the extracellular data are reasonably similar but vary somewhat from each other (Table 2 and Table 3).

However, the β2AR intracellular face shows significant differences between receptors in different groups. As shown in Figure 3b, not only are there visible variations between structures, the active and inactive groups separate entirely. This also occurs with alignment to the PC1 axis which could therefore describe β2AR activation; a negative PC1 score correlates with inactive structures (red), while a positive score correlates with active (blue and green). The intracellular ANOSIM and PERMANOVA data also gave relatively higher R and F values (0.8967 and 16.17, *p* = 0.0001 and *p* = 0.0001) which is indicative of dissimilarity between groups (Table 2 and Table 3). Interestingly, while overlapping with the inactive group (red), the active (blue) and active coupled to G_s_ (green) groups were seen to be distinct at the extracellular face, but not intracellularly.

These analyses have therefore clearly demonstrated a difference between the active and inactive β2AR structures at the intracellular face of the transmembrane helix bundle, as hypothesised. These results were also supported by the significance of the statistical tests and provide proof of concept that the application of geometric morphometrics with principal component analysis to GPCRs was successful and could also be applied to further receptor characteristics as well.

### 2.3. Bound Ligands—The Family B Receptors

Whilst differences were observed with receptor activation state, these are often caused by the ligands bound to them. Just as there are a wide variety of GPCRs, their ligands are just as diverse, including light, ions, small molecules, peptides and proteins; each of these can be categorised by the effect they exert on the receptors and its pharmacology. These include agonist, partial agonist, biased agonist, antagonist, inverse agonist and positive and negative allosteric modulators (PAM and NAM) [6]. It was therefore hypothesised that variation would be observed in GPCR structures with these different categories of ligand bound to them. The family B receptor group was selected due to ample numbers and a mixture of ligand-bound structures. Of the 54 family B structures studied, 38 were described as agonist-bound, one partial agonist-bound, six antagonist-bound, five as negative allosteric modulator-bound and three in an unbound state. A list of the PDBs is provided in Appendix A
Table A2. Structure 6FJ3 was not included in analyses due to unusable annotation and 6ORV was not included due to PDB unavailability.

After PC scoring, agonist-bound structures were labelled red, partial agonist blue, antagonist yellow, NAM green and unbound purple (Figure 4). In a similar way to the activation states of the β2-adrenergic receptors, the extracellular face of the family B structures showed variation, with agonist bound structures largely in the negative distribution of data; however, again, there was a general overlap and no clear clustering (Figure 4a). ANOSIM and PERMANOVA gave low R and F values (0.01442 and 3.02, *p* = 0.4207 and *p* = 0.0024) which indicates similarity between these groups (Table 4 and Table 5). This makes sense biologically as the family B receptors often share affinity to several of their peptide hormone ligands, albeit to a varying extent.

However, again, there were more visible differences at the intracellular face of family B (Figure 4b). Similarly, the ligand categories aligned to the PC1 axis showing a gradient of active to inactive effects from negative to positive. Agonist-bound structures (red) correlated with a negative PC1 score and less active or inactive structures gave a positive score. Unbound structures fell somewhere between the extremes of agonist and antagonist/NAM. ANOSIM and PERMANOVA R and F values were relatively higher for the intracellular face (0.7606 and 13.94, *p* = 0.0001 and *p* = 0.0001) indicative of dissimilarity between these ligand groups (Table 4 and Table 5).

These results therefore show shape variation between ligand-bound receptors within family B, at the intracellular face, again supported by the statistical data. These data perhaps show less clear separation between bound ligand types due to family B being larger, and containing several sub-groups within it, when compared to the β2AR group. Furthermore, many of the family B receptors often co-express with an accessory protein needed for trafficking or full activation [27]; a lack of which may hinder their true structural dynamics. Nevertheless, there are clear groupings, demonstrating the applicability of geometric morphometric analysis in different scenarios.

### 2.4. Fusion Proteins—The Orexin Receptors

Fusion proteins can be incorporated into the target receptor for a variety of reasons including greater stability during expression, a cleavable target during purification and enhanced identification during crystallisation. Glycogen synthase is one such fusion protein, in this case replacing part of ICL3 of the orexin receptors. This was engineered to enhance crystallisation during the vapor diffusion and lipidic cubic phase methods of X-ray diffraction [28]. Note that each of these receptors had first undergone a series of 12 thermostabilising mutations to create the StaR (stabilised receptor) proteins [29], as well as the fusion. It was hypothesised that adding a 196 amino acid insertion into ICL3 may have an impact on the orexin receptor structures, undetectable through pharmacology. It must be remembered that crystallography provides a ‘snapshot’ structure and, although proteins may function in cells where they are dynamic, their determined structures could be biased towards particular conformations by the introduction of such fusions; this is an important consideration in the field. Indeed, to our knowledge, these structures have not been compared previously and geometric morphometric analysis will enable quantification of any differences due to fusions. Of the 22 orexin receptors studied, nine contained the fusion protein and 13 did not. A list of the PDBs is provided in Appendix A
Table A3.

Following PC scoring, glycogen synthase fusions were labelled red, and no fusion labelled as blue (Figure 5). The extracellular face data showed the receptors with fusion proteins clustered in the bottom-right space of the graph with some overlap of the non-fusion group (Figure 5a). ANOSIM and PERMANOVA was performed, resulting in low R and F values (0.1018 and 4.33, *p* = 0.0879 and *p* = 0.0152) indicating similarity between groups (Table 6 and Table 7).

The intracellular face was observed to largely separate the two groups aligned to the PC1 axis (Figure 5b). Glycogen synthase fusion correlated with the positive PC score, with non-fusion as the negative. ANOSIM and PERMANOVA R and F values were higher than the extracellular data (0.6271 and 16.73, *p* = 0.0001 and *p* = 0.0002) indicating more dissimilarity at the intracellular face (Table 6 and Table 7). PC1, representing the largest variation in structure, could therefore correlate with the structural alterations caused by inserting glycogen synthase into ICL3. Note that the variation was much clearer at the intracellular face which was the location of the fusion protein.

There was one anomalous result observed where an orexin fusion (red) was found with the non-fusion group (blue) in the intracellular data. Nonetheless, we believe this is the first demonstration of geometric morphometrics in being able to significantly classify GPCR structures based upon the presence or absence of a glycogen synthase fusion protein.

### 2.5. Thermostabilised Receptors

During the expression, purification and experimentation of proteins, it is common practice to keep samples on ice as much as possible to prevent degradation of the sample’s quality or structural integrity [16]. This can be especially challenging with regards to techniques used for solving GPCR structures and so the introduction of mutations can improve the receptor’s stability at higher temperatures [30]. This is often achieved by systematic mutagenesis which ultimately increases the protein’s rigidity, decreases overall mobility and gives more stable enthalpy stemming from improved interhelical packing when compared to wildtype [30]. These mutations have been shown to cause no pharmacological differences in some cases, but can in others; for example, some mutations in the neurotensin 1 receptors (NTS1R) cause them to no longer signal at all [31]. It is therefore important to understand whether these mutations cause significant structural changes or not, and whether they are ‘safe’ to use when determining GPCR structures. Several examples of receptor families were selected as they are well studied and include ample numbers of thermostabilised structures to compare against the non-mutated receptors. This group included the adenosine A2a receptors (A2AR), the β1-adrenergic receptors (β1AR) and the β2-adrenergic receptors (β2AR). In total, these groups included 60 thermostabilised structures and 64 non-mutated. A list of the PDBs is provided in Appendix A
Table A1, Table A4 and Table A5.

After PC scoring, thermostabilised structures were labelled red, and non-mutated as blue (Figure 6). Overall, the data analysed showed no significant differences between the thermostabilised and non-mutated receptors, at both the extra- and intracellular faces of the TM bundle. When analysing this larger groups of receptors, extracellular ANOSIM and PERMANOVA tests gave low R and F values (0.07405 and 5.442, *p* = 0.0003 and *p* = 0.0024) indicating similarity between groups (Table 8 and Table 9). Similarly, the intracellular R and F values were also low (0.02588 and 2.854, *p* = 0.0452 and *p* = 0.0525), indicative of similarity as well (Table 8 and Table 9). This observation was also consistent with other analyses performed (data not shown). When the thermostabilised data in Figure 6 were separated between the A2AR, β1AR and β2AR receptors, or including other GPCRs (such as the neurotensin and substance P receptors), results were consistent, and no significant differences were observed in any analysis.

These results suggest that, in terms of the receptors analysed, thermostabilising mutations did not cause any significant variation from the non-mutated receptors. This supports the idea of utilising thermostabilising mutations to enhance the experimentation of GPCRs and, possibly, other important membrane proteins as well. This reassurance is particularly valuable as many medically relevant GPCRs can often undergo these mutations, such as the StaR engineering process [29], before application to clinical research/studies.

## 3. Discussion

GPCRs are widely studied membrane protein receptors due to their involvement in disease states, and potential as drug targets. Despite their importance, development of novel, specific and effective prophylaxis can often be hindered by a lack of structural knowledge and understanding of these dynamic receptors. Thus, the application of innovative techniques to this field will undoubtedly further enhance understanding of GPCR structures and modifications, to ultimately improve their druggability. We hypothesised that differences between GPCR structures, under varying conditions, could be detected, quantified and analysed statistically using geometric morphometrics. This mathematical technique was therefore used with principal component analysis to assess receptor shape variation in a three-dimensional, unbiased manner. The results obtained demonstrated that this technique was successfully applied to GPCR structures and gave insight into the modifications researchers frequently perform with them.

As geometric morphometrics is mainly utilised in disciplines such as anthropology and palaeontology, it was first necessary to understand if it could be applied to protein structures as well. One unpublished example exists on the bioRxiv database which examines the soluble α-amylase enzymes with geometric morphometrics [32]. However, after an initial pilot study and further expansion of our methods, we believe this is the first application of this technique to GPCRs, and membrane proteins in general, especially on this scale.

We first demonstrated that the XYZ coordinates of amino acids are not only compatible with the MorphoJ software, but produce figures which make sense. The generalised Procrustese transformations presented (e.g., Figure 2a) show the average helical positions which, when compared back to GPCR structures, accurately represent a view of the transmembrane helix bundle at both the extra- and intracellular faces. This meant that the coordinate data was transformed into a standardised and reliable dataset from which the subsequent analyses could be performed with confidence. Similarly, the overall direction and magnitude of the dataset’s variation was able to be portrayed by the lollipop graphs (e.g., Figure 2b), indicating which helices vary the most under different conditions. Finally, the principal component score comparisons (e.g., Figure 2d) provided an unbiased platform to then infer meaning by grouping receptors together and statistically analysing the dissimilarity between those groups.

Initially, receptor activation was selected as the characteristic to provide proof of concept as it is the main source of natural variation between structures. It is a movement native to all wild-type GPCRs, and is reasonably well understood [33]. Based on the generally accepted model of activation, it was hypothesised that GPCRs resolved in different activation states should vary from one another. Helical movement would therefore be expected, with a focus on the intracellular face of the transmembrane bundle. Reassuringly, these predictions were proved to be correct as shown in Appendix B
Figure A1b,e. TM6 was highlighted as the main location of variation between activation states, moving away from the TM bundle as expected. Subsequently, comparison of the intracellular principal component scores (Figure 3b) showed very clear separation between activation states. The group of inactive β2AR receptors were observed to significantly differ from the active or active with G_s_ groups, and thus further confirm our hypothetical predictions. These results are therefore likely indicative of the conformational movement to allow post-activation binding of the intracellular G protein and propagation of the appropriate signalling pathway, as predicted. Additionally, the significant separation of the active and active coupled to G_s_ groups at the extracellular face should also be noted (Figure 3a). This unexpected result suggests that the binding of a stimulatory G protein possibly causes bias to the extracellular face of the β2AR TM bundle to adopt a different conformation than those with only agonist bound to them. Interestingly, this was not observed at the intracellular face despite being the location of G protein binding, providing a thought-provoking avenue for further research. Indeed, it has been observed that binding of a G protein to a GPCR can increase the affinity for ligands, which may be as a result of changes to the extracellular face [34]. Nevertheless, these convincing results provide strong evidence to support the use of geometric morphometrics with GPCRs and that it can be further applied to highlight structural variation under additional conditions.

As receptor activation largely relies upon ligand binding, this was the next characteristic studied. This is especially interesting as GPCRs can interact with a variety of different ligand types, and with bias towards certain ligands or signalling pathways. One explanation may involve a specific helical movement which opens an intracellular binding cleft biased towards certain G proteins, either stimulatory or inhibitory, etc. We therefore hypothesised that GPCR structures with different ligand classes bound to them may vary from each other, again, possibly focussed at the intracellular face. The extracellular data of the family B receptors showed a general similarity with no significantly different groupings (Figure 4a). This possibly suggests that as the N terminus of their peptide hormone ligands bind to a residue close to the TM bundle and ECL2, the extracellular face of the TM bundle arrangement remains more similar across the family B receptors, perhaps due to their shared ligand affinity [35]. However, the intracellular face exhibited significant variation (Figure 4b), with agonist bound structures shown to be most dissimilar to the structures resolved with antagonist or NAM bound to them, and unbound structures falling somewhere between the two. One possible explanation for this may include a greater involvement of the intracellular helix arrangement in the bias towards certain G protein pathways, as a result of ligand binding. It is also interesting to observe that the structural variation of the partial agonist-bound receptor (blue) was more similar to antagonist-bound rather than agonist-bound. Looking in more depth at this PDB file (5YQZ), it is nonetheless described as inactive which could explain this observation [36]. Furthermore, several of these family B receptors can require co-expression with an accessory protein called a receptor activity modifying protein (RAMP) [27]. Biologically, these RAMPs can influence trafficking to the cell membrane, glycosylation and receptor pharmacology. Crystallisation without a RAMP, in some cases, could therefore hinder a family B GPCR’s full dynamic range, and should be considered when studying their resolved structures. Nonetheless, these results demonstrate the ability of geometric morphometrics to significantly discriminate structures based upon their bound ligands, with a meaningful gradient of activation across the principal component score axis.

While the analysis of activity and bound ligands supported our hypotheses based on existing GPCR understanding, we also focussed our attention to modifications frequently performed during the experimentation of proteins. These included the insertion of an intracellular fusion protein and the introduction of receptor thermostabilising mutations, highlighting some important results to share.

Firstly, insertion of a glycogen synthase fusion protein into the ICL3 of the orexin receptors caused significant variation at the intracellular face (Figure 5b). Despite seemingly having no effect on their pharmacology, the presence of the fusion protein appears to largely bias these receptors to a significantly different intracellular conformation. This is important for the field to be aware of during experimental design as intracellular fusion proteins could sterically hinder the receptor’s full range of motion or bias structures to more stable conformations that do not necessarily fully represent the in vivo state of the protein. One anomalous result was found to have the opposite effect where the insertion of a fusion protein caused no difference from those without the fusion. It is interesting that this structure (6V9S) was bound with a novel antagonist (JH112) made by enantiospecific synthesis [37], whereas the other structures had their typical non-selective antagonist Suvorexant bound to them. JH112 is a sub-nanomolar antagonist to activate G_q_ mediated recruitment of β-arrestin to the orexin 1 receptors [37], and further study is likely required to confirm whether JH112 can bias the orexin receptor to a different intracellular conformation than the other antagonist-bound structures. In any case, we believe this is the first demonstration of using geometric morphometrics to indicate that an intracellular fusion modification can cause significant variation at the intracellular face of GPCRs.

Finally, thermostabilising mutations remain to be a more controversial area of GPCR research as conflicting opinions on its use still persist. Some groups advocate for the advantages thermostability provides experimentally, especially when resolving difficult structures [38]. However, others promote a degree of caution when using non-native proteins and emphasise the need to ensure proteins are active. Either way, thermostabilising mutations remain a prevalent component of the holistic approach to studying protein structures. Ultimately, our analyses did not detect any significant differences between thermostabilised and non-mutated GPCRs, from a variety of receptor sub-groups (Figure 6); this perhaps suggests that, on a geometric level, thermostabilising mutations are “safe” to use. However, given that some receptors are no longer able to signal after this modification, it would be pragmatic to comprehensively analyse one’s GPCR of interest, nonetheless.

One downside to this method is its manual nature; the variety of annotated PDB files makes automation difficult to attempt though this may become possible in the future. Despite this, geometric morphometrics is an overwhelmingly advantageous technique. Landmark coordinates are not restricted to the ends of the transmembrane helices, other locations such as the middle of helices could be analysed, and they must simply be common to each structure. Similarly, this technique is not limited to GPCRs and could be applied to other proteins of interest such as ABC transporters and aquaporins [39,40]. While interesting results and structural differences were observed with regards to the characteristics presented in this paper, we have also been able to classify receptors based on further factors including organism species, highlighting possible future analyses. With regards to GPCRs, an area requiring further study is the explanation of G protein coupling specificity [41], especially concerning intermediate states and the role of the intracellular binding cleft. Another exciting possibility involves checking the validity of the recent AlphaFold project predictions—at the time of writing, these predicted structures seem reasonable although limited to inactive states for now [42,43]. To conclude, the results presented in this paper provide a proof of concept for the use of geometric morphometrics in the study of GPCR structures, especially when variation may be undetectable by pharmacological assays or other structural techniques such as root-mean-square deviation (RMSD). We have therefore adapted a method to apply this mathematical technique and demonstrate meaningful and statistically significant analyses. It is ultimately intended to be a tool to aid sense-checking newly resolved structures and planning early experimental design, even beyond GPCRs.

## 4. Materials and Methods

A database of all currently known GPCR structures was first obtained from the *GPCRdb* website (https://gpcrdb.org/structure/ (accessed on 21 August 2021)) and cross-referenced to the *mpstruc* database (https://blanco.biomol.uci.edu/mpstruc/ (accessed on 21 August 2021)) to ensure the working model was based upon a comprehensive list of resolved structures (last sampled 30 June 2021) [26,44]. From here, the PDB code of each structure was used to download the .pdb file from the Research Collaboratory for Structural Bioinformatics Protein Data Bank (https://www.rcsb.org/ (accessed on 21 August 2021)). Next, the Swiss-PdbViewer (DeepView) software version 4.10 (http://www.expasy.org/spdbv/ (accessed on 21 August 2021)) was used to visualise and manipulate the structures before data collection [8]. Finally, data processing and statistical analysis was performed with the MorphoJ software (https://morphometrics.uk/MorphoJ_page.html (accessed on 21 August 2021)) and the PAST data analyser software (https://past.en.lo4d.com/windows (accessed on 21 August 2021)) [24,45]. All of these are freely available.

Each receptor structure was analysed using the same process as follows. Firstly, the amino acid residues located at the very end of each transmembrane helix were identified manually and the name and sequence number recorded. As GPCRs have seven transmembrane helices, this results in fourteen landmarks for each structure: Seven at the extracellular face, and seven at the intracellular face. Next, the .pdb file was opened as text and the x, y and z coordinates located and recorded for each of the fourteen identified residues. Specifically, these were the coordinates for the alpha-carbon atom (Cα) for each residue. These were then divided into two groups (extra- and intracellular) and then exported as .txt files. This manual process was repeated exactly for every structure analysed. We envisage that this process could be automated, although the disparity between presentation and annotation of the .pdb files could well hinder attempts.

A new project was then started in MorphoJ, with the relevant .txt file imported as three-dimensional data (the extracellular data was processed separately to the intracellular). Two preliminary processes were then performed. Firstly, a Procrustese fit was performed with the data aligned by principal axes, and then a covariance matrix was generated. Finally, principal component analysis was performed which gives the principal component shape changes, eigenvalues and scores [24]. The PC score comparisons were then used to identify groups, clusters and outliers based upon the characteristics of the GPCR structure. Any overlap or differences in data have been highlighted by the addition of coloured convex hulls to the PC comparisons which demarcate the smallest area containing each group.

These PC scores were then tested statistically using PAST [45]; one-way permutational multivariate analysis of variance (PERMANOVA) tested the distance between the centroids of each group, while one-way analysis of similarity (ANOSIM) tested group dissimilarity. Both of these multivariate tests involved 9999 permutations and pairwise comparisons were performed using Euclidean distances. PERMANOVA gives P and F values, testing for differences between groups by distance; a larger F value indicates a more pronounced group separation. ANOSIM gives P and R values, using the mean rank of distances between and within groups; an R value of 1 indicates complete dissimilarity. Significance for both tests is calculated by 10,000 permutations of group membership [17].

## 5. Conclusions

Historically, GPCR structures have been typically difficult to study, owing to their complex, dynamic structures, and the limitations of membrane protein production at the time. As these processes have developed and improved over the last few decades, so have the quantity and quality of resolved GPCR structures; as of September 2021, there are over one thousand GPCR structure models available, with nearly two hundred thousand ligands and thirty thousand ligand interactions discovered so far (GPCR*db*). This puts the field in a position to discover potential treatments for the many conditions which still have limited or no options in terms of effective prophylaxis involving GPCRs. We have first demonstrated that the novel application of geometric morphometrics with principal component analysis to GPCRs was successful as a proof of concept, and is able to discriminate between structures based on their characteristics. This method can therefore be used as a tool to further study the variation between structures, correlated with these characteristics including receptor activity, bound ligand classes, presence of fusion proteins etc. Notably, our results have thus far suggested that thermostabilising mutations do not cause significant differences. Overall, we believe this is a beneficial, unbiased technique with wide applicability to the structural biology field and can form the basis for further development of geometric morphometrics with GPCRs.

## Figures and Tables

**Figure 1 pharmaceuticals-14-00953-f001:**
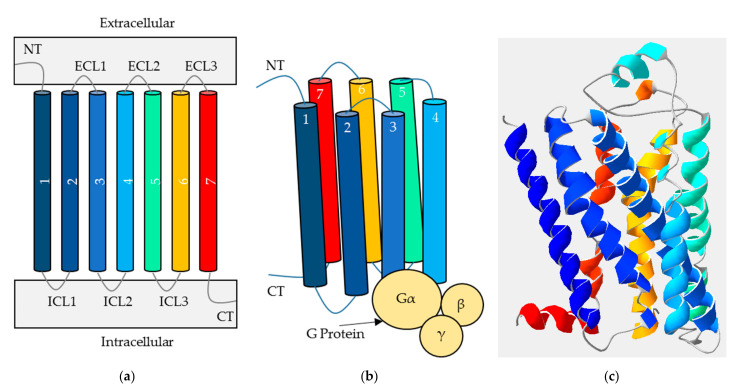
The typical structure of a G protein-coupled receptor (GPCR). (**a**) A two-dimensional representation of a GPCR showing its extracellular N terminus (NT), seven transmembrane helices (TM1-7, multicoloured), three extracellular loops (ECL1-3), three intracellular loops (ICL1-3) and intracellular C terminus (CT). (**b**) A three-dimensional representation of the transmembrane bundle, with helices labelled 1–7 and intracellular recruitment of a G protein (yellow circles). (**c**) An example of a resolved GPCR crystal structure for context (β2AR receptor, PDB code 6PS0) [8].

**Figure 2 pharmaceuticals-14-00953-f002:**
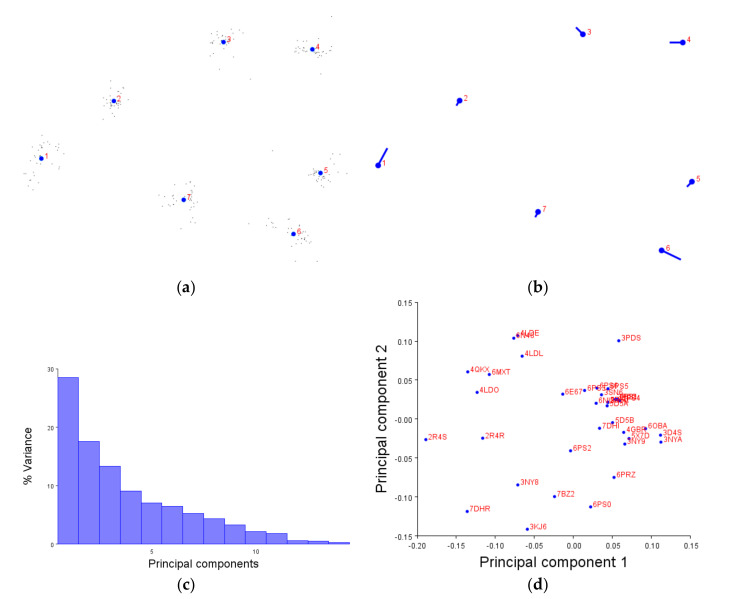
The four graphs generated by geometric morphometrics and principal component analysis using extracellular β2AR data as an example. (**a**) Generalised Procrustese transformation standardises the dataset to show an average position (blue numbered dots) and the individual data (black dots) around them. In this case, this can be thought of as a top-down view of the extracellular TM bundle. (**b**) Lollipop graph visualising morphospace variation caused by principle component 1 (PC1). The average position of each helix is represented by the blue dot; the stick protruding away from the dot shows the direction and magnitude of variation caused by PC1 (the greatest source of variation). (**c**) Principal component eigenvalues and the percentage of variance they account for—PC1 always gives the most variance, and then PC2 second, and so on. (**d**) Comparison between PC1 and PC2; this is used to assess the greatest variation of the dataset’s morphospace and to observe patterns produced by the data. Each dot represents a GPCR structure within the selected dataset and can be labelled with PDB codes or different colours. A list of structures included in each figure can be found in the Appendix A tables, and further supplementary figures in Appendix B.

**Figure 3 pharmaceuticals-14-00953-f003:**
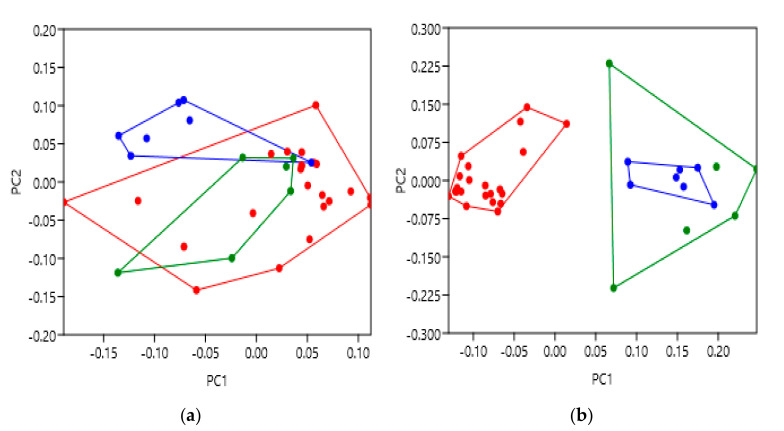
Principal component analysis to show the β2-adrenergic receptor (β2AR) shape variation between activation states. Inactive structures are red, active are blue and active with G_s_ coupled are green. (**a**) PC score comparison for the extracellular face of the β2AR transmembrane bundle. (**b**) PC score comparison for the intracellular face of the β2AR transmembrane bundle. The supporting statistical data are summarised in Table 2 and Table 3. Additional supporting figures are found in Appendix B
Figure A1.

**Figure 4 pharmaceuticals-14-00953-f004:**
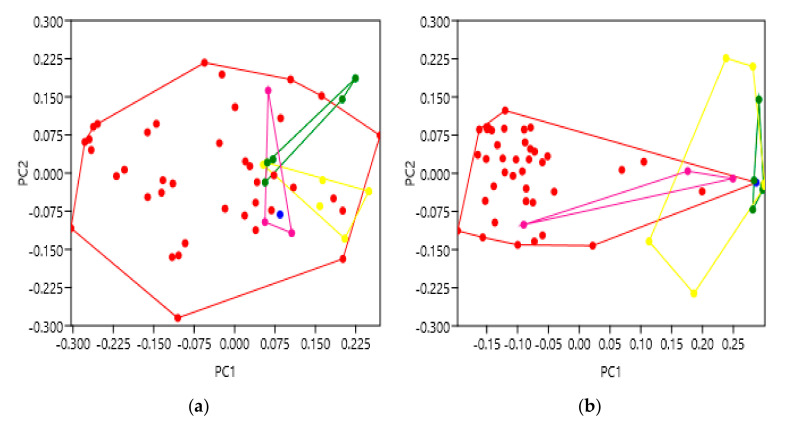
Principal component analysis to show the family B receptor shape variation due to bound ligands. Agonist-bound structures are red, antagonist yellow, negative allosteric modulator (NAM) green, partial agonist blue and unbound purple. (**a**) PC score comparison for the extracellular face of the family B transmembrane bundle. (**b**) PC score comparison for the intracellular face of the family B transmembrane bundle. The supporting statistical data are summarised in Table 4 and Table 5. Additional supporting figures are found in Appendix B Figure A2.

**Figure 5 pharmaceuticals-14-00953-f005:**
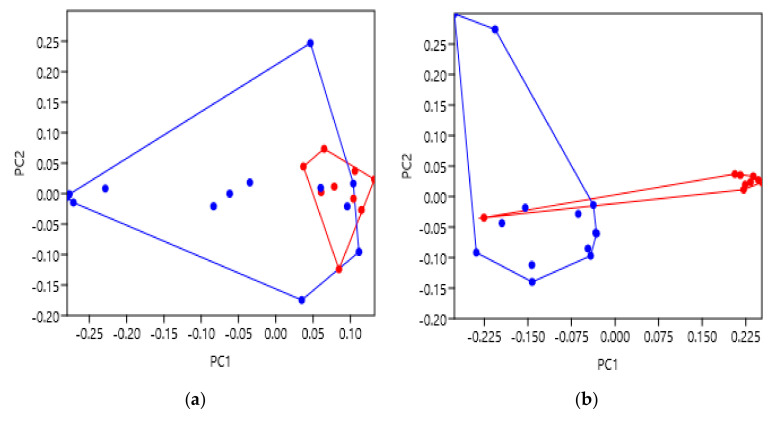
Principal component analysis to show the orexin receptor shape variation with a glycogen synthase fusion protein or not. Those with a glycogen synthase fusion are indicated in red, and those with no fusion are blue. (**a**) PC score comparison for the extracellular face of the orexin receptors transmembrane bundle. (**b**) PC score comparison for the intracellular face of the orexin receptors transmembrane bundle. The supporting statistical data are summarised in Table 6 and Table 7. Additional supporting figures are found in Appendix B Figure A3.

**Figure 6 pharmaceuticals-14-00953-f006:**
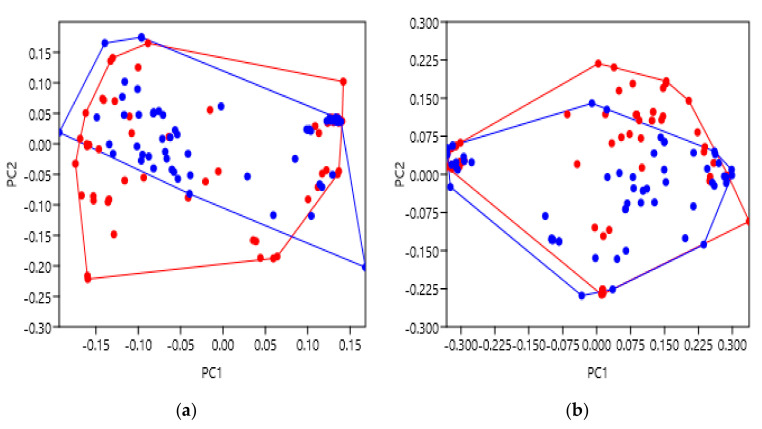
Principal component analysis to show the shape variation between non-mutated and thermostabilised receptors. Non-mutated are blue and thermostabilised are red. (**a**) PC score comparison for the extracellular face of the A2AR, β1AR and β2AR transmembrane bundles. (**b**) PC score comparison for the intracellular face of the A2AR, β1AR and β2AR transmembrane bundles. The supporting statistical data are summarised in Table 8 and Table 9. Additional supporting figures are found in Appendix B Figure A4.

**Table 1 pharmaceuticals-14-00953-t001:** A classification summary of the six G protein-coupled receptor (GPCR) families [7]. GPCRs are typically grouped based on sequence homology and functional similarity, though many are still considered ‘orphans’—receptors of unknown function or classification. Family D and E (found in fungi and slime moulds) are not understood as well as the other four families, due to their limited application to human prophylaxis; mammalian receptors have understandably taken priority with structural studies and pharmaceutical research.

Receptor Family	Description	Features	Ligand Examples
A	Rhodopsin-like	Lacks a substantial extracellular domain; thus native ligands bind directly to the transmembrane domain	Light, small molecules, peptides, proteins (e.g., opioids, vasopressin, neurotensin, epinephrine, etc.)
B	Secretin-like	Flexible, folded extracellular domain containing a hormone binding motif	Peptide hormones (e.g., calcitonin, glucagon, parathyroid hormone, etc.)
C	Metabotropic glutamate receptors	Dynamic extracellular domain containing a venus flytrap module	Ions and small molecules (e.g., gamma-amino butyric acid, glutamate, calcium, etc.)
D	Fungal mating pheromone receptors	Activates MAPK signals to induce cell-cell fusion forming a diploid zygote	Cell membrane mating factors (alpha-factor and a-factor)
E	cAMP receptors	Binds DNA	Cyclic AMP
F	Frizzled/Smoothened receptors	Beginning of Wnt and Hedgehog signalling pathways	Wnt protein ligands, Smoothened agonist (SAG), cholesterol

**Table 2 pharmaceuticals-14-00953-t002:** A summary of ANOSIM (clustering) test data for the extra- and intracellular faces of the β2AR receptors, to support Figure 3; 9999 permutations performed on PC1-5, significant *p* values (<0.05) are shown in bold.

Extracellular				Intracellular			
ANOSIM				ANOSIM			
R value		0.2965		R value		0.8967	
*p* value		0.0047		*p* value		0.0001	
**Pairwise *p* Values**				**Pairwise *p* Values**			
	Inactive	Active	G_s_		Inactive	Active	G_s_
Inactive		**0.0026**	0.1273	Inactive		**0.0001**	**0.0001**
Active	**0.0026**		**0.0073**	Active	**0.0001**		**0.0062**
G_s_	0.1273	**0.0073**		G_s_	**0.0001**	**0.0062**	

**Table 3 pharmaceuticals-14-00953-t003:** A summary of PERMANOVA (distance) test data for the extra- and intracellular faces of the β2AR receptors, to support Figure 3; 9999 permutations performed on PC1-5, significant *p* values (<0.05) are shown in bold.

Extracellular				Intracellular			
PERMANOVA				PERMANOVA			
F value		4.142		F value		16.17	
*p* value		**0.0003**		*p* value		**0.0001**	
**Pairwise *p* Values**				**Pairwise *p* Values**			
	Inactive	Active	G_s_		Inactive	Active	G_s_
Inactive		**0.0005**	0.1354	Inactive		**0.0001**	**0.0001**
Active	**0.0005**		**0.0166**	Active	**0.0001**		0.0753
G_s_	0.1354	**0.0166**		G_s_	**0.0001**	0.0753	

**Table 4 pharmaceuticals-14-00953-t004:** A summary of ANOSIM (clustering) test data for the extra- and intracellular faces of the family B receptors, to support Figure 4. For statistical analysis, the partial agonist structure (5YQZ) was included in the agonist group; 9999 permutations performed on PC1-5, significant *p* values (<0.05) are shown in bold.

Extracellular					Intracellular				
ANOSIM					ANOSIM				
R value		0.01442			R value		0.7606		
*p* value		**0.4207**			*p* value		**0.0001**		
**Pairwise *p* Values**					**Pairwise *p* Values**				
	Agonist	Antagonist	NAM	Unbound		Agonist	Antagonist	NAM	Unbound
Agonist		0.4181	0.1937	0.5556	Agonist		**0.0001**	**0.0001**	**0.0132**
Antagonist	0.4181		0.1211	0.2648	Antagonist	**0.0001**		0.3958	0.2146
NAM	0.1937	0.1211		0.3855	NAM	**0.0001**	0.3958		**0.0352**
Unbound	0.5556	0.2648	0.3855		Unbound	**0.0132**	0.2146	**0.0352**	

**Table 5 pharmaceuticals-14-00953-t005:** A summary of PERMANOVA (distance) test data for the extra- and intracellular faces of the family B receptors, to support Figure 4. For statistical analysis, the partial agonist structure (5YQZ) was included in the agonist group; 9999 permutations performed on PC1-5, significant *p* values (<0.05) are shown in bold.

Extracellular					Intracellular				
PERMANOVA					PERMANOVA				
F value		3.02			F value		13.94		
*p* value		**0.0024**			*p* value		**0.0001**		
**Pairwise *p* Values**					**Pairwise *p* Values**				
	Agonist	Antagonist	NAM	Unbound		Agonist	Antagonist	NAM	Unbound
Agonist		**0.0054**	**0.0059**	0.3527	Agonist		**0.0001**	**0.0001**	**0.0105**
Antagonist	**0.0054**		**0.0246**	0.7125	Antagonist	**0.0001**		0.3886	0.1684
NAM	**0.0059**	**0.0246**		0.1635	NAM	**0.0001**	0.3886		**0.0171**
Unbound	0.3527	0.7125	0.1635		Unbound	**0.0105**	0.1684	**0.0171**	

**Table 6 pharmaceuticals-14-00953-t006:** A summary of ANOSIM (clustering) test data for the extra- and intracellular faces of the orexin receptors, to support Figure 5; 9999 permutations performed on PC1-5, significant *p* values (<0.05) are shown in bold.

Extracellular			Intracellular		
ANOSIM			ANOSIM		
R value	0.1018		R value	0.6271	
*p* value	0.0879		*p* value	**0.0001**	
**Pairwise *p* Values**			**Pairwise *p* Values**		
	Glycogen synthase	No fusion		Glycogen synthase	No fusion
Glycogen synthase		0.0825	Glycogen synthase		**0.0001**
No fusion	0.0825		No fusion	**0.0001**	

**Table 7 pharmaceuticals-14-00953-t007:** A summary of PERMANOVA (distance) test data for the extra- and intracellular faces of the orexin receptors, to support Figure 5; 9999 permutations performed on PC1-5, significant *p* values (<0.05) are shown in bold.

Extracellular			Intracellular		
PERMANOVA			PERMANOVA		
F value	4.33		F value	16.73	
*p* value	**0.0152**		*p* value	**0.0002**	
**Pairwise *p* Values**			**Pairwise *p* Values**		
	Glycogen synthase	No fusion		Glycogen synthase	No fusion
Glycogen synthase		**0.0134**	Glycogen synthase		**0.0001**
No fusion	**0.0134**		No fusion	**0.0001**	

**Table 8 pharmaceuticals-14-00953-t008:** A summary of ANOSIM (clustering) test data for the extra- and intracellular faces of the A2AR, β1AR and β2AR receptors, to support Figure 6; 9999 permutations performed on PC1-5, significant *p* values (<0.05) are shown in bold.

Extracellular			Intracellular		
ANOSIM			ANOSIM		
R value	0.07405		R value	0.02588	
*p* value	**0.0003**		*p* value	**0.0452**	
**Pairwise *p* Values**			**Pairwise *p* Values**		
	Thermostabilised	Non-mutated		Thermostabilised	Non-mutated
Thermostabilised		**0.001**	Thermostabilised		**0.0456**
Non-mutated	**0.001**		Non-mutated	**0.0456**	

**Table 9 pharmaceuticals-14-00953-t009:** A summary of PERMANOVA (distances) test data for the extra- and intracellular faces of the A2AR, β1AR and β2AR receptors, to support Figure 6; 9999 permutations performed on PC1-5, significant *p* values (<0.05) are shown in bold.

Extracellular			Intracellular		
PERMANOVA			PERMANOVA		
F value	5.442		F value	2.854	
*p* value	**0.0024**		*p* value	0.0525	
**Pairwise *p* Values**			**Pairwise *p* Values**		
	Thermostabilised	Non-mutated		Thermostabilised	Non-mutated
Thermostabilised		**0.0018**	Thermostabilised		0.0564
Non-mutated	**0.0018**		Non-mutated	0.0564	

## Data Availability

Data is available via Aston Research Explorer (https://research.aston.ac.uk/; https://doi.org/10.17036/researchdata.aston.ac.uk.00000521 (accessed on 21 August 2021)).

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
