# Peer review of "The Novel Application of Geometric Morphometrics with Principal Component Analysis to Existing G Protein-Coupled Receptor (GPCR) Structures"

_pharmaceuticals, 2021, doi:10.3390/ph14100953_

Round 1

Reviewer 1 Report

This article is the first demonstration of Geometric Morphometrics with Principal Component Analysis  suitability for membrane G-protein analysis.
This elaboration especially succes due to use of new sophisticatedmorphometric methodology. It is a very well-described and documented article. It written in clear and transparent language.

  1.    In the abstract a few words about used methodology should be included 
    2.    In the Introduction examples of some very well-known GPER in various organ/tissue and their role together with GPER mechanisms of actions should be provided

3. Findings by Prossnitz group should be helpful

Author Response

This article is the first demonstration of Geometric Morphometrics with Principal Component Analysis  suitability for membrane G-protein analysis. This elaboration especially success due to use of new sophisticated morphometric methodology. It is a very well-described and documented article. It written in clear and transparent language.

  1. In the abstract a few words about used methodology should be included: We have included this suggestion in the abstract.
  2. In the Introduction examples of some very well-known GPER in various organ/tissue and their role together with GPER mechanisms of actions should be provided: The GPER receptors are not related to this paper’s topic and we have therefore chosen not to include them.
  3. Findings by Prossnitz group should be helpful: The Prossnitz group studies the GPER receptors, and we have therefore not referenced this group.

Reviewer 2 Report

Manuscript ID: Pharmaceuticals 1369025

In the presented article, using databases with available  structures of the G protein-coupled receptors (GPCR), the authors demonstrate that geometric morphometrics methodology can be useful to detected, quantify and analyze receptor shape variation in a 3-dimensional manner. Scientific approach is very interesting, however, some corrections have to be applied to make the manuscript more understandable for Pharmaceuticals readers.

My critical remarks concern mainly not clear statement of the aim in the abstract, not precise descriptions in the introduction sections, results elucidations and presentation. Manuscript should be corrected therefore, my recommendation for it is MAJOR REVISION.

COMMENTS:

  • Please modify the abstract section to include the statement what was the aim of the study
  • The message from the body text in lines 44-93, Figure 1 and Table 1 is not clear and seem to be not related with mathematical approach authors want highlight as a good tool to understand better/describe better the GPCR structure. Please improve description by e.g. providing information which information is necessary to use geometric morphometrics to analyze GPCR; why the presented information do not help to understand functional meaning of known data about GPCR structure.
  • The role of figure 1 (especially “b” and “c”) is not to illustrate three domains structure of GPCRs, therefore, please move the Figure 1 reference from line 44 to more suitable place or rephrase the sentence where you introduce the reference of Fig. 1.
  • Please improve appearance of Fig 1a and Fig 1b for better understanding of the message of Fig. 1. I recommend: in (a) highlight by different fold or colors 3 parts (domains) of GPCR; in (b) please add names of helices TM1-TM7; in (a) and (b) please change colors of transmembrane helices to make them easy comparable with Fig. 1c; in (b) please move green triangle to make illustration consistent with the text - In lines 48-49 you mention that orthosteric ligand binding place is in N terminus (not in ECL1 as indicated in Fig 1b); in Figure 1 caption is not clear reference for Fig 1c information “rendered in Swiss PDB Viewer” is confusing and not related to presented in Fig 1 data- please remove unnecessary text.
  • Classical system of classification of GPCRs is not clear from the manuscript. In most cases authors say about six “families” of GPCR but sometimes (e.g. in Table 1 caption) use “class”. Please improve consistency and in name of column “Classification” add appropriate systematic nomenclature
  • Data presented in Table 1 are laconic. Please improve and complete description to make these data more informative. E.g. please specify what do you mean by words “short”(1x), “small” (2x), “rich” respectively in phrases: short N terminus, small molecules, cysteine rich. Moreover, please specify the meaning “similar to class A”. Additionally, give specific examples of ligands in column “ligand examples”. Also did you mean >Wnt signaling pathways< by the phrase “Wnt proteins”?

Text in lines 50-52 describing of Table 1 is confusing. Did you mean that names of six GPCR families are A-F?. Please rephrase sentence

  • Sentence in lines 76-77 is confusing and suggests that all G proteins (also small G proteins) are composed of three subunits. Please correct.
  • Line 81. Please rephrase the sentence or explain better what do you mean by “cell behavior”.
  • In paragraph (lines 103-116) please specify what do you mean by “each end of helix”. Do you mean EC1-EC3?
  • In Fig 2b and Figs in Appendix B1-B4 in “b” and “e” there is no scale for “stick protruding”. Please add the scale to understand magnitude of variation caused by PC1
  • The body text of the section 2 is hard to understand because authors combined here description of obtained results and methodological approach. I recommend move methodological descriptions to section 4 (Materials and methods) and leave in section 2 only original data authors present in the manuscript. When necessary you can send the readers to Material and Method section for better understanding
  • Once entered, the abbreviation should be used consistently. I found that the abbreviations and full names are used in the article inappropriately. Sometimes authors use abbreviations and full names by chance: e.g. for PC, PC1, PC2; for ANOSIM; PERMANOVA introduce abbreviations in place where abbreviations is not mentioned for the first time. In some cases introducing abbreviations seem to be not necessary (e.g. PCA in line 552);sometimes introduced abbreviations are not explained (e.g. RMSD in line 521). Please improve consistency in the usage of abbreviations throughout the manuscript. A list of abbreviations could help the understanding of the text.

Author Response

  1. Please modify the abstract section to include the statement what was the aim of the study: A statement of our aims has been included in the abstract.
  2. The message from the body text in lines 44-93, Figure 1 and Table 1 is not clear and seem to be not related with mathematical approach authors want highlight as a good tool to understand better/describe better the GPCR structure. We have altered figure 1 to aid in understanding of the structure of GPCRs.
  3. Please improve description by e.g. providing information which information is necessary to use geometric morphometrics to analyze GPCR; Later in the introduction, we describe the data required for this analysis “Geometric morphometrics can therefore quantifiably capture and display these shape variations between resolved GPCR structures, in a three-dimensional format. The data required for this method are Cartesian landmark coordinates and should be common reference points between the shapes [18]. As such, the seven transmembrane helices of GPCRs can be used as they are the defining structural characteristic of these receptors and remain relatively conserved between receptors.”
  4. why the presented information do not help to understand functional meaning of known data about GPCR structure. We have revisited this section in light of the reviewer’s comments.  However, we find that the sections are clear and relevant.  It is important to highlight the different classes and activation mechanisms of GPCRs as these are investigated in the results section.  If we have misunderstood the referee then we are happy to revisit these sections.
  5. The role of figure 1 (especially “b” and “c”) is not to illustrate three domains structure of GPCRs, therefore, please move the Figure 1 reference from line 44 to more suitable place or rephrase the sentence where you introduce the reference of Fig. 1: Figure 1 has been updated so this is no longer inappropriate.
  6. Please improve appearance of Fig 1a and Fig 1b for better understanding of the message of Fig. 1. I recommend: in (a) highlight by different fold or colors 3 parts (domains) of GPCR; in (b) please add names of helices TM1-TM7; in (a) and (b) please change colors of transmembrane helices to make them easy comparable with Fig. 1c; in (b) please move green triangle to make illustration consistent with the text - In lines 48-49 you mention that orthosteric ligand binding place is in N terminus (not in ECL1 as indicated in Fig 1b); in Figure 1 caption is not clear reference for Fig 1c information “rendered in Swiss PDB Viewer” is confusing and not related to presented in Fig 1 data- please remove unnecessary text: Figure 1a and 1b have been updated to include colours matching Figure 1c. Transmembrane helices have been labelled 1-7. The ligand (green triangle) has been removed to prevent confusion and unnecessary text removed as well.
  7. Classical system of classification of GPCRs is not clear from the manuscript. In most cases authors say about six “families” of GPCR but sometimes (e.g. in Table 1 caption) use “class”. Please improve consistency and in name of column “Classification” add appropriate systematic nomenclature: The word family has been used more consistently, to replace the word class where appropriate.
  8. Data presented in Table 1 are laconic. Please improve and complete description to make these data more informative. E.g. please specify what do you mean by words “short”(1x), “small” (2x), “rich” respectively in phrases: short N terminus, small molecules, cysteine rich. Moreover, please specify the meaning “similar to class A”. Additionally, give specific examples of ligands in column “ligand examples”. Also did you mean >Wnt signaling pathways< by the phrase “Wnt proteins”?: Table 1 has been updated to include more detail and explanation.
  9. Text in lines 50-52 describing of Table 1 is confusing. Did you mean that names of six GPCR families are A-F?. Please rephrase sentence: Rephrased for clarity.
  10. Sentence in lines 76-77 is confusing and suggests that all G proteins (also small G proteins) are composed of three subunits. Please correct: This has been corrected.
  11. Line 81. Please rephrase the sentence or explain better what do you mean by “cell behavior”: Explanation included for clarity.
  12. In paragraph (lines 103-116) please specify what do you mean by “each end of helix”. Do you mean EC1-EC3?: Included an explanation for more clarity.
  13. In Fig 2b and Figs in Appendix B1-B4 in “b” and “e” there is no scale for “stick protruding”. Please add the scale to understand magnitude of variation caused by PC1: A scale would not be appropriate to include as the protrusions are relative to each other, within each figure; this is standard practice within the field of geometric morphometics.
  14. The body text of the section 2 is hard to understand because authors combined here description of obtained results and methodological approach. I recommend move methodological descriptions to section 4 (Materials and methods) and leave in section 2 only original data authors present in the manuscript. When necessary you can send the readers to Material and Method section for better understanding: As we are presenting the proof of concept of a novel method, we believed that this was an appropriate way of allowing readers to understand the method and results. We would therefore prefer to keep this section as it is.
  15. Once entered, the abbreviation should be used consistently. I found that the abbreviations and full names are used in the article inappropriately. Sometimes authors use abbreviations and full names by chance: e.g. for PC, PC1, PC2; for ANOSIM; PERMANOVA introduce abbreviations in place where abbreviations is not mentioned for the first time. In some cases introducing abbreviations seem to be not necessary (e.g. PCA in line 552);sometimes introduced abbreviations are not explained (e.g. RMSD in line 521). Please improve consistency in the usage of abbreviations throughout the manuscript. A list of abbreviations could help the understanding of the text: Abbreviations have been defined with their first use, and updated to be used more consistently throughout.

Round 2

Reviewer 2 Report

The authors addressed all my critical remarks and applied necessary corrections to the current version of the manuscript, which improved its quality. The revised version of the Article: “The Novel Application of Geometric Morphometrics with Principal Component Analysis to Existing G Protein-Coupled Receptor (GPCR) Structures” by Daniel N. Wiseman et al., in my opinion, is ready for publication in Pharmaceuticals.